# Sounding Bodies: Modeling 3D Spatial Sound of Humans Using Body Pose and Audio

**Xudong Xu**[*]
Shanghai AI Laboratory
xudongxu9710@gmail.com

**Dejan Marković**
Meta Reality Labs Research
dejanmarkovic@meta.com

**Jacob Sandakly**
Meta Reality Labs Research
jasandakly@meta.com

**Todd Keebler**
Meta Reality Labs Research
toddkeebler@meta.com

**Steven Krenn**
Meta Reality Labs Research
stevenkrenn@meta.com

**Alexander Richard**
Meta Reality Labs Research
richardalex@meta.com

## Abstract

While 3D human body modeling has received much attention in computer vision, modeling the acoustic equivalent, *i.e.* modeling 3D spatial audio produced by body motion and speech, has fallen short in the community. To close this gap, we present a model that can generate accurate 3D spatial audio for full human bodies. The system consumes, as input, audio signals from headset microphones and body pose, and produces, as output, a 3D sound field surrounding the transmitter's body, from which spatial audio can be rendered at any arbitrary position in the 3D space. We collect a first-of-its-kind multimodal dataset of human bodies, recorded with multiple cameras and a spherical array of 345 microphones. In an empirical evaluation, we demonstrate that our model can produce accurate body-induced sound fields when trained with a suitable loss. Dataset and code are available online.[1]

## 1   Introduction

Throughout all of our history, humans have been at the center of the stories we tell: starting from early cave paintings, over ancient tales and books, to modern media such as movies, TV series, computer games, or, more recently, virtual and augmented realities. Not surprisingly, digital representations of the human body have therefore long been a research topic in our community Guan et al. [2009], Loper et al. [2015], Kanazawa et al. [2018]. With the rise of computer-generated imagery in the film industry and the availability of powerful computing on consumer devices, visual representations of 3D human bodies have seen great advances recently, leading to extremely lifelike and photorealistic body models Bagautdinov et al. [2021], Zheng et al. [2023].

However, while the visual representation of human bodies has made steady progress, its acoustic counterpart has been largely neglected. Yet, human sensing is intrinsically multi-modal, and visual and acoustic components are intertwined in our perception of the world around us. Correctly modeling 3D sound that matches the visual scene is essential to the feeling of presence and immersion in a

---

[*]Work done during internship at Meta Reality Labs Research, Pittsburgh, PA, USA.
[1]https://github.com/facebookresearch/SoundingBodies

37th Conference on Neural Information Processing Systems (NeurIPS 2023).

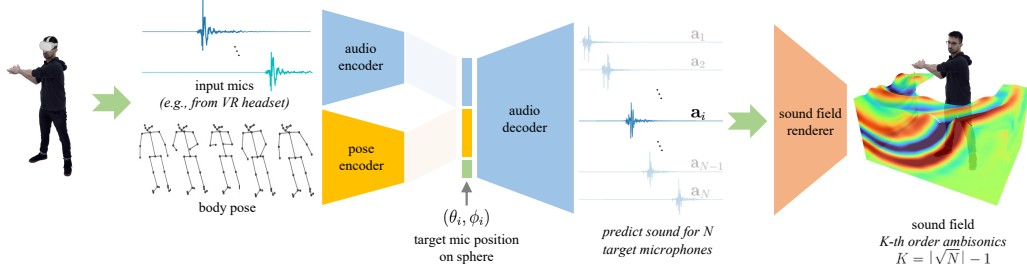

Figure 1: **System Overview.** The model consumes audio from microphones on a headset and body pose and encodes them into latent audio and pose embeddings. An audio decoder that is conditioned on a target microphone position then generates an audio signal as it would sound from the given target position. Note that the decoder can only render target microphones that lie on a sphere around the transmitter as this is the capture setup for the training data. In order to model the full sound field at any arbitrary position, we predict the signal at $N = (K + 1)^2$ microphone positions on a sphere surrounding the body and compute the $K$-th order ambisonic sound field from these signals.

3D environment Hendrix and Barfield [1996]. In this work, we propose a method to close this gap between the visual and acoustic representation of the human body. We demonstrate that accurate 3D spatial sound can be generated from head-mounted microphones and human body pose.

In particular, we consider an AR/VR telepresence scenario where people interact as full-body avatars. In this setting, available inputs include egocentric audio data from the head-mounted microphones (it is common for headsets to employ microphone arrays), as well as body pose used to drive the avatar, obtained either using external sensors or from the headset itself (the current generation of headsets are able to track the upper body / hands). The goal is to model the user-generated sound field so that it be correctly rendered at arbitrary positions in the virtual space.

Existing approaches to sound spatialization, both in traditional signal processing Savioja et al. [1999] and in neural sound modeling Richard et al. [2021, 2022], operate under the strict assumption that the sound to be spatialized has a known location and is also recorded at this location, without interference through other sounds. These assumptions are in stark contrast with the task we address in this work. First, the location of the sound is unknown, *i.e.*, a finger snap could come from the right hand as well as the left hand. Second, the sound can not be recorded at the location where it is produced. In particular, the microphones are typically head-mounted (as in an AR/VR device), such that finger snaps or footsteps are not recorded at the location of their origin. Third, it is usually not possible to record a clean signal for each sound source separately. Body sounds like footsteps or hand-produced sounds often co-occur with speech and are therefore superimposed in the recorded signal.

To overcome these challenges, we train a multi-modal network that leverages body pose to disambiguate the origin of different sounds and generate the correctly spatialized signal. Our method consumes, as input, audio from seven head-mounted microphones and body pose. As output, the sound field surrounding the body is produced. In other words, our method is able to synthesize 3D spatial sound produced by the human body from only head-mounted microphones and body pose, as shown in Figure 1.

Our major contributions are (1) a novel method that allows rendering of accurate 3D sound fields for human bodies using head-mounted microphones and body pose as input, (2) an extensive empirical evaluation demonstrating the importance of body pose and of a well-designed loss, and (3) a first-of-its-kind dataset of multi-view human body data paired with spatial audio recordings from an array with 345 microphones. We recommend watching the supplemental video before reading the paper.

## 2 Related Work

**Audio-Visual Learning.** In our physical world, audio and video are implicitly and strongly connected. By exploiting the multi-modal correlations, plenty of self-supervised learning methods are proposed for representation learning and significantly outperform those approaches learning from a single modality Owens et al. [2016], Korbar et al. [2018], Alwassel et al. [2020], Tian et al. [2020], Xiao et al.

[2020], Patrick et al. [2021], Morgado et al. [2020, 2021a,b]. Besides, such a close cross-modality correlation is also shown to boost the performance in audio-visual navigation Chen et al. [2020, 2021], Gan et al. [2020, 2022], speech enhancement Owens and Efros [2018], Yang et al. [2022], audio source localization Qian et al. [2020], Hu et al. [2020], Tian et al. [2021], Jiang et al. [2022], Mo and Morgado [2022] and separation Afouras et al. [2018], Ephrat et al. [2018], Zhao et al. [2018], Gao et al. [2018], Gao and Grauman [2021]. Among these works, the most related papers to ours focus on audio-driven gesture synthesis Ginosar et al. [2019], Lee et al. [2019], Ahuja et al. [2020], Yoon et al. [2020], which aims to generate a smooth sequence of the human skeleton coherent with the given audio. While Ginosar et al. [2019], Ahuja et al. [2020], Yoon et al. [2020] focus on the translation from speech to conversational gestures based on the cross-modal relationships, Lee et al. [2019] explore a synthesis-by-analysis learning framework to generate a dancing sequence from the input music. Different from them, we are interested in the opposite direction, *i.e.*, reconstructing the 3D sound field around the human from the corresponding body pose.

**Visually-guided Audio Spatialization.** Visually-guided audio spatialization is to transform monaural audio into 3D spatial audio under the guidance of visual information. There is an increasing number of data-driven methods for this task, where crucial spatial information is obtained from the corresponding videos. Morgado et al. [2018], Li et al. [2018] and Huang et al. [2019] leverage $360°$ videos to generate the binaural audio and concurrently show the capability of localizing the active sound sources. Besides, Gao and Grauman [2019] regards the mixture of left and right channels as pseudo mono audio and utilizes the visual features to predict the left-right difference with a learnable U-Net. After that, a line of papers Lu et al. [2019], Yang et al. [2020], Zhou et al. [2020], Garg et al. [2021], Xu et al. [2021] follow a similar framework and achieve better spatialization quality with different novel ideas. However, these methods fail to model time delays and reverberation effects, making the binaural audio less realistic. As for this problem, Richard et al. [2021] propose to spatialize speech with a WaveNet van den Oord et al. [2016] conditioned on the source and listener positions explicitly. Despite the remarkable progress in audio spatialization, these approaches are all limited to binaural audio or first-order ambisonics and thus fail to recover the whole sound field. In contrast, our method strives to generate the whole 3D sound field by using the corresponding 3D human body pose.

## 3 Dataset

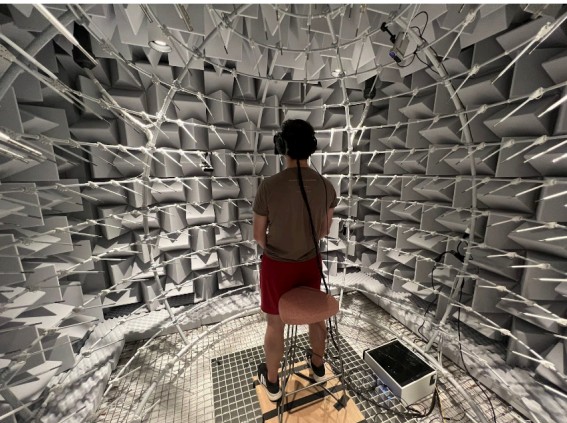

Figure 2: Capture stage with five body-tracking cameras and 345 microphones on a sphere around the participant.

**Setting.** In this work we focus on modeling speech and body generated sounds considering an AR/VR telepresence scenario. In general, it is desirable to model these sounds in an anechoic environment, *i.e.*, without acoustics of a specific room, which allows to add arbitrary room reverberations using traditional signal processing methods (*e.g.*, convolution with room impulse responses). For this reason we collect the dataset in an anechoic chamber. We note that the user's real environment can be noisy and reverberant, which could affect the input audio signals from the head-mounted microphones. There is a huge corpus of research around denoising and dereverberation and we are not addressing these problems in this paper.

**Data capture.** We capture a corpus of paired audio-visual data in an anechoic chamber with 5 Kinects for body tracking and 345 microphones arranged on a sphere around the center of the capture stage. The cameras capture at 30 frames per second and audio is recorded at 48kHz. The Kinects and 345 microphones are co-calibrated such that they share the same world coordinate system. During the capture, the participants are standing in the center of the stage while wearing a head-mounted device with seven microphones to gather input audio.

In total, we collect 4.4h of human body sounds and speech from eight different participants. We capture each participant in three different settings: standing and wearing light clothes like a T-Shirt that does not produce a lot of noise while moving; wearing heavy clothes like jackets or coats that

produce audible noise while moving; and sitting in light clothing. For each setting, participants are instructed to perform a variety of body sounds such as clapping, snapping, tapping their body, footsteps, or scratching themselves, as well as speaking. The full capture script can be found in the supplemental material. Overall, 50% of the data are non-speech body sounds and 50% are speech. Note that handling both kinds of data poses an additional challenge for our approach: speech is typically a dense and structured signal, while body sounds are sparse, impulsive, and contain rich high-frequency components.

**Data processing.** We extract body poses from each camera using OpenPose Cao et al. [2017] and obtain 3D keypoints via triangulation of the poses from the five camera views. The audio signals from all microphones (345 microphones on the spherical array and seven head-mounted microphones) and the Kinects are time-synchronized using an IRIG time code. Technical details about audio-visual time synchronization can be found in the supplemental material.

## 4 Pose-Guided 3D Spatial Audio Generation

Producing full 3D sound fields inherently requires multimodal learning from both audio input and pose input, as neither modality alone carries the full spatial sound information. We formalize the problem and describe the system outlined in Figure 1, with a focus on network architecture and loss that enable effective learning from the information-deficient input data.

### 4.1 Problem Formulation

Let $\mathbf{a}_{1:T} = (a_1, \ldots, a_T)$ be an audio signal of length $T$ captured by the head-mounted microphones, where $T$ indicates the number of samples, and each $a_t \in \mathbb{R}^{C_{\text{in}}}$ represents the value of the waveform at time $t$ for each of the $C_{\text{in}}$ input channels. Let further $\mathbf{p}_{1:S} = (p_1, \ldots, p_S)$ be the temporal stream of body pose, where each $p_s \in \mathbb{R}^{J \times 3}$ represents the 3D coordinates of each of the $J$ body joints at the visual frame $s$. Note that audio signals $\mathbf{a}_{1:T}$ and pose data $\mathbf{p}_{1:S}$ are of the same duration, but sampled at a different rate. In this work, audio is sampled at 48kHz and pose at 30 fps, so there is one new pose frame for every 1,600 audio samples.

Ideally, we would like to learn a transfer function $\mathcal{T}$ that maps from the head-mounted microphone signals and body pose directly to the audio signal $\mathbf{s}_{1:T}$ at any arbitrary coordinate $(x, y, z)$ in the 3D space,

$$\mathbf{s}_{1:T}(x, y, z) = \mathcal{T}(x, y, z, \mathbf{a}_{1:T}, \mathbf{p}_{1:S}). \tag{1}$$

However, it is in practice impossible to densely populate a 3D space with microphones and record the necessary training data. Instead, our capture stage is a spherical microphone array and data can only be measured on the surface of this sphere, *i.e.* at microphone positions defined by their polar coordinates $(r, \theta, \phi)$. Since the radius $r$ is the same for every point on the sphere, a microphone position is uniquely defined by its azimuth $\theta$ and elevation $\phi$.[2] Leveraging the recordings of the microphones on the sphere as supervision, we model a transfer function

$$\mathbf{s}_{1:T}(\theta_i, \phi_i) = \mathcal{T}(\theta_i, \phi_i, \mathbf{a}_{1:T}, \mathbf{p}_{1:S}) \tag{2}$$

that maps from input audio $\mathbf{a}_{1:T}$ and body pose $\mathbf{p}_{1:S}$ to the signal recorded by microphone $i$ which is located at $(\theta_i, \phi_i)$ on the sphere.

In order to render sound at an arbitrary position $(x, y, z)$ in 3D space, we predict $\mathbf{s}_{1:T}(\theta, \phi)$ for all microphone positions $(\theta_i, \phi_i)$ on the sphere, and encode them into harmonic sound field coefficients Samarasinghe and Abhayapala [2012], from which the signal $\mathbf{s}_{1:T}$ can then be rendered at any arbitrary spatial position $(x, y, z)$, see Section 4.4 for details.

### 4.2 Network Architecture

In this section, we describe how we implement the transfer function defined in Equation (2). At the core of the model are an audio encoder and a pose encoder which project both input modalities onto a latent space, followed by an audio decoder that has a WaveNet van den Oord et al. [2016] like

---

[2]This is true for a perfectly spherical microphone array. In practice, the microphones are mounted imperfectly and slight deviations from the surface of the sphere occur, which we neglect here.

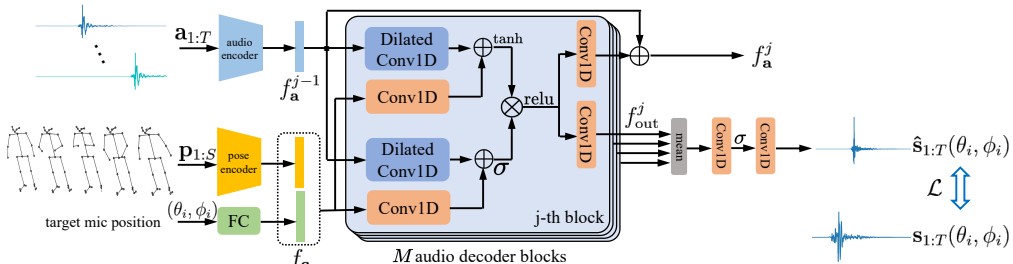

Figure 3: **Network Architecture.** The model consists of an encoder for each of the input modalities and a stack of $M$ WaveNet-like audio decoding blocks. In the $j$-th block, the input are the conditioning features $f_\mathbf{c}$ and the audio encoding $f_\mathbf{a}$ ($j = 1$) or the output of the previous block $f_\mathbf{a}^{j-1}$ ($j > 1$), respectively. The final output is an average-pooled accumulation of all block's outputs $f_{\text{out}}^j$, forwarded through two convolutional layers.

architecture and generates the target sound $\mathbf{s}_{1:T}$ at a given microphone position $(\theta_i, \phi_i)$. The model architecture is outlined in Figure 3.

As training data, we use recordings from the capture stage described in Section 3, where we have fully synchronized and paired data of headset input, body pose, and target microphones. In other words, our training data are tuples $(\mathbf{a}_{1:T}, \mathbf{p}_{1:S}, \mathbf{s}_{1:T}(\theta_i, \phi_i))$ for each microphone position $(\theta_i, \phi_i)$ on the spherical array.

**Audio Encoder.** Acoustic propagation through space at the speed of sound causes a time delay between the emitting sound source (*e.g.*, a hand), the recording head-mounted microphones, and the actual target microphone on the spherical microphone array. It has been found beneficial to apply *time-warping*, *i.e.*, to shift the input signal by this delay, before feeding it into a network Richard et al. [2021]. In our case, as sound might emerge from various locations on the body, we create a copy of the input signal for multiple body joints and shift the input as if it originated from each of these joints. We then concatenate all these shifted input copies along the channel axis and feed them into a linear projection layer to obtain the audio encoding $f_\mathbf{a}$ that contains information about the time-delayed audio signal from all possible origins, see the supplemental material for details.

**Pose Encoder.** While a person is making some sound, their body poses $\mathbf{p}_{1:S}$ give strong cues *when* and *where* the sound is emitted. Additionally, sound interacts with the body and propagates differently through space depending on body pose. Hence, the signal $\mathbf{s}_{1:T}$ at a target microphone position $(\theta_i, \phi_i)$ depends on the body pose at that time. For the input pose sequence $\mathbf{p}_{1:S}$, the 3D coordinates of each body joint are first encoded to latent features and the temporal relation is aggregated via two layers of temporal convolutions with kernel size 5. Then, we concatenate all body joint features and use an MLP to fuse them into a compact pose feature $f_\mathbf{p}$.

**Target Microphone Position.** In addition to audio and pose encodings, the decoder depends on the position $(\theta_i, \phi_i)$ of the target microphone. We first map the microphone location into a continuous Cartesian space

$$(x, y, z) = \big( \cos(\theta_i) \sin(\phi_i), \sin(\theta_i) \sin(\phi_i), \cos(\phi_i) \big)$$

and project it to a higher-dimensional embedding using a learnable, fully-connected layer.

Pose encoding and target microphone position encoding are then concatenated into a conditioning feature vector $f_\mathbf{c}$ which is upsampled to 48kHz and, together with the warped input audio, passed to the audio decoder.

**Audio Decoder.** Following prior works van den Oord et al. [2016], Richard et al. [2021], we stack $M$ WaveNet-like spatial audio generation blocks as illustrated in Figure 3, in which the input audio features $f_\mathbf{a}$ are transformed into the desired output signal $\mathbf{s}_{1:T}(\theta_i, \phi_i)$ under the guidance of the conditioning pose and position features $f_\mathbf{c}$. Each block is a gated network combining information from audio and conditioning features. In the $j$-th block, the audio feature $f_\mathbf{a}^j$ passes through a dilated Conv1D layer and is added to conditioning features processed by Conv1D layers in both filter and gate branches, which are then integrated after the $\tanh$ and sigmoid activation functions:

$$\mathbf{z} = \tanh(W_f^j * f_\mathbf{a}^j + V_f^j * f_\mathbf{c}) \odot \sigma(W_g^j * f_\mathbf{a}^j + V_g^j * f_\mathbf{c}), \tag{3}$$

where $*$ denotes a convolution operator, $\odot$ indicates element-wise multiplication, $\sigma$ is a sigmoid function, $j$ is the index of the decoder block, $f$ and $g$ denote filter and gate, respectively, and $W, V$ are both learnable convolution filters. A ReLU activation is applied to $\mathbf{z}$ which is then processed by two separated Conv1D layers. One layer predicts the residual audio feature to update $f_{\mathbf{a}}^{j}$ to $f_{\mathbf{a}}^{j+1}$ for the next generation block, and another is responsible for producing the output audio feature $f_{\text{out}}^{j}$. All output features $f_{\text{out}}^{1,\dots,M}$ from $M$ blocks are average pooled and then decoded to raw audio waves $\hat{\mathbf{s}}_{1:T}(\phi_i, \theta_i)$ via two Conv1D layers. Finally, for training, the predicted audio is compared with the ground-truth recordings $\mathbf{s}_{1:T}(\phi_i, \theta_i)$ using loss functions $\mathcal{L}$ (see Section 4.3).

### 4.3 Loss Function

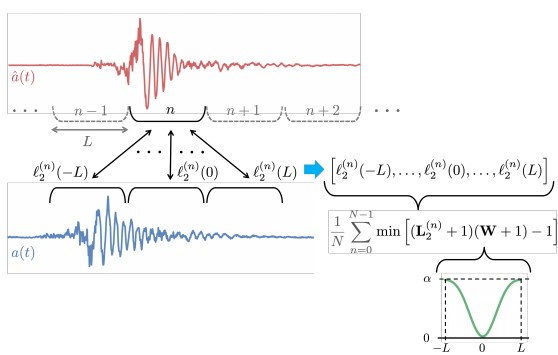

Optimizing for correct spatialization of body sounds is challenging due to both the nature of the task (correct spatialization requires correct time alignment), and the nature of the signals we are dealing with (a mix of dense and structured speech with sparse, impulsive sounds such as clapping and snapping). To avoid speech distortions, a loss on the amplitude spectrogram is generally preferred. On the other hand, such loss is unable to achieve accurate temporal alignment. Alternatively, the time domain $\ell_2$ loss has been successfully used for neural synthesis of binaural speech [Richard et al. [2021]].

Figure 4: Illustration of the shift-$\ell_2$ loss computation.

However, in our experiments we found that $\ell_2$ tends to highly attenuate impulsive sounds. One issue is that small shifts of impulsive signals cause big spikes in $\ell_2$ error in combination with the fact that available information is not sufficient for a perfect sample-level alignment (body keypoints give only a rough indication of a possible origin of sound). To address this issue we propose a variation of $\ell_2$ that is more forgiving to small shift offsets. The loss, referred here a shift-$\ell_2$, is computed over sliding windows with higher penalty applied to higher offsets.

We start by dividing the estimated signal $\hat{a}(t)$ into $N$ segments of size $L$ as illustrated in Figure 4. For each segment $n$, we compute a weighed $\ell_2$ error against a sliding window from a reference signal $a(t)$, with sliding offset $\tau$ going from $-L$ and $L$. The error is computed as

$$\ell_2^{(n)}(\tau) = \frac{1}{L} \sum_{t=1}^{L} \left| \frac{\hat{a}(nL+t) - a(nL+t+\tau)}{\sqrt{\sigma_a \min(\sigma_a, \sigma_{\hat{a}})} + \delta} \right|^2 , \tag{4}$$

where $\sigma$ indicates the standard deviation of the signal, and $\delta$ is small value used to avoid numerical issues. The result is a set of $2L+1$ values $\mathbf{L}_2^{(n)} = \left[ \ell_2^{(n)}(-L), \dots, \ell_2^{(n)}(0), \dots, \ell_2^{(n)}(L) \right]$ representing $\ell_2$ errors normalized by the signals energy and computed over a set of shift offsets. After penalizing larger offsets by using a window function $\mathbf{W} = \alpha \left( 1 - [w(-L), \dots, w(0), \dots, w(L)] \right)$, with $w(\tau)$ denoting the Blackman window of length $2L+1$, and $\alpha$ being the penalty coefficient, we select the minimum value as $\min \left[ (\mathbf{L}_2^{(n)} + 1)(\mathbf{W} + 1) - 1 \right]$. By design this value can be 0 only if the $\ell_2$ error is 0 at shift $\tau = 0$. Finally, the loss value is obtained by averaging the results of each of $N$ segments

$$\text{shift-}\ell_2 = \frac{1}{N} \sum_{n=0}^{N-1} \min \left[ (\mathbf{L}_2^{(n)} + 1)(\mathbf{W} + 1) - 1 \right] . \tag{5}$$

We use L = 128, $\alpha$ = 100 , $\delta$ = 0.001. To reduce the distortions of speech data, we combine shift-$\ell_2$ with a multiscale STFT loss [Yamamoto et al. [2021]] computed for window sizes of 256, 128, 64, 32.

### 4.4 Sound Field Rendering

Given the audio signals at each microphone and the knowledge of microphone locations, we encode the 3D sound field in the form of harmonic sound field coefficients following [Samarasinghe and Abhayapala [2012]]. The maximum harmonic order $K$ is limited by the number of microphones $N$ as $K = \lfloor \sqrt{N} \rfloor - 1$. For our system with 345 microphones this allows sound field encoding up to 17-th harmonic order. More details can be found in the supplemental material.

Table 1: The role of body pose is critical to solving the task. Both, on speech and non-speech signals, we see significant improvements when the model has access to the full body pose as opposed to no pose or headpose only.

| | non-speech | | | | speech | | |
|---|---|---|---|---|---|---|---|
| pose information | SDR ↑ | amplitude ↓ | phase ↓ | | SDR ↑ | amplitude ↓ | phase ↓ |
| no pose | 2.518 | 0.823 | 0.464 | | 6.682 | 9.007 | 1.362 |
| headpose only | 2.848 | 0.809 | 0.461 | | 9.264 | 7.093 | 1.190 |
| full body pose | **3.004** | **0.784** | **0.454** | | **9.580** | **7.059** | **1.184** |

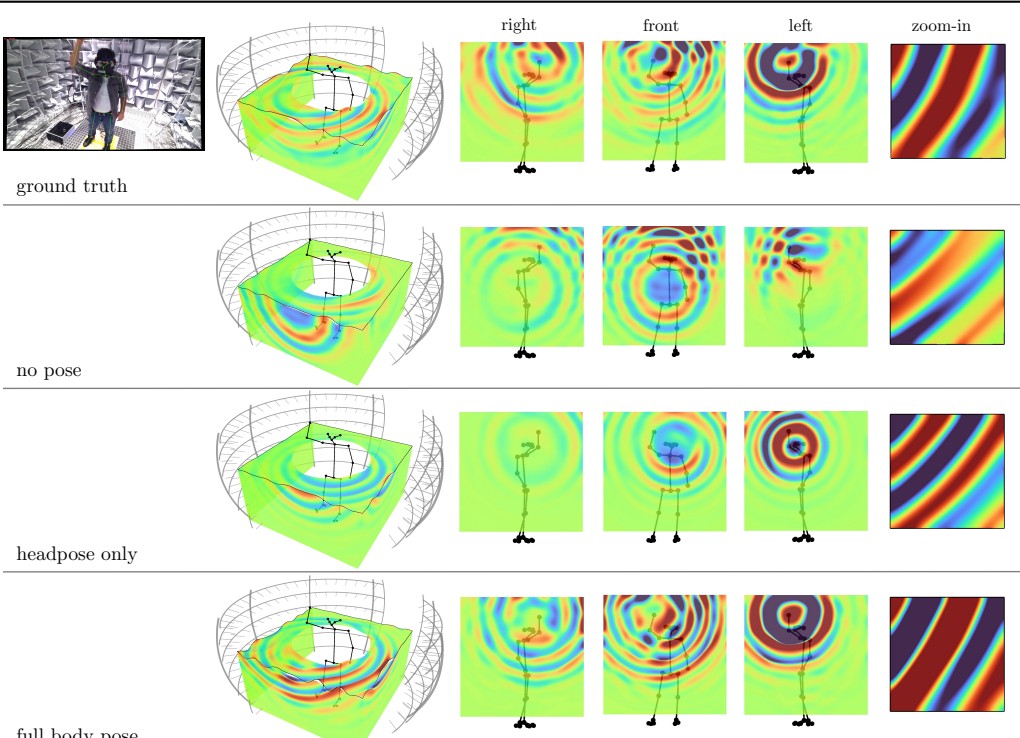

Figure 5: Without pose, the audio signal always originates from the center of the 3D space. With only headpose, the model always predicts the head as the sound source location. Only when provided full body pose, the model is able to correctly localize the sound source.

Given the encoded harmonic sound field coefficients $\beta_{nm}(\tau, f)$, where $\tau$ and $f$ are, respectively, time and frequency bins, the sound pressure at any point $\boldsymbol{x} = (r, \theta, \phi)$ in space[3] can be decoded as Williams [1999]

$$p(\boldsymbol{x}, t) = \text{iSTFT}\left( \sum_{n=0}^{K} \sum_{m=-n}^{n} \beta_{nm}(\tau, f) h_n(kr) Y_{nm}(\theta, \phi) \right) , \qquad (6)$$

where $k = 2\pi f / v_{\text{sound}}$ is the wave number, $v_{\text{sound}}$ is the speed of sound; $Y_{nm}(\theta, \phi)$ represents the spherical harmonic of order $n$ and degree $m$ (angular spatial component), and $h_n(kr)$ is $n$th-order spherical Hankel function (radial spatial component).

## 5    Experiments

**Dataset.** All the experiments are conducted on our collected dataset. We cut the recordings into clips of one second in length without overlapping, resulting in 15,822 clips in total. The dataset is partitioned into train/validation/test sets of 12,285/1,776/1,761 clips, respectively. The test set consists of 961 non-speech clips and 800 speech clips. To the best of our knowledge, this is the first

---

[3]we use polar coordinates here instead of Cartesian coordinates for simpler notation in Equation (6).

Table 2: Besides pose, multiple head-mounted microphones provide a source of spatial information. Increasing the number of microphones from one to seven leads to substantial improvements in output quality. Note that all experiments here use full body pose.

| # input mics | non-speech | | | speech | | |
|---|---|---|---|---|---|---|
| | SDR ↑ | amplitude ↓ | phase ↓ | SDR ↑ | amplitude ↓ | phase ↓ |
| 1 | 0.848 | 0.849 | 0.478 | 8.259 | 7.956 | 1.235 |
| 3 | 2.338 | 0.812 | 0.458 | 9.526 | 7.276 | 1.199 |
| 7 | **3.004** | **0.784** | **0.454** | **9.580** | **7.059** | **1.184** |

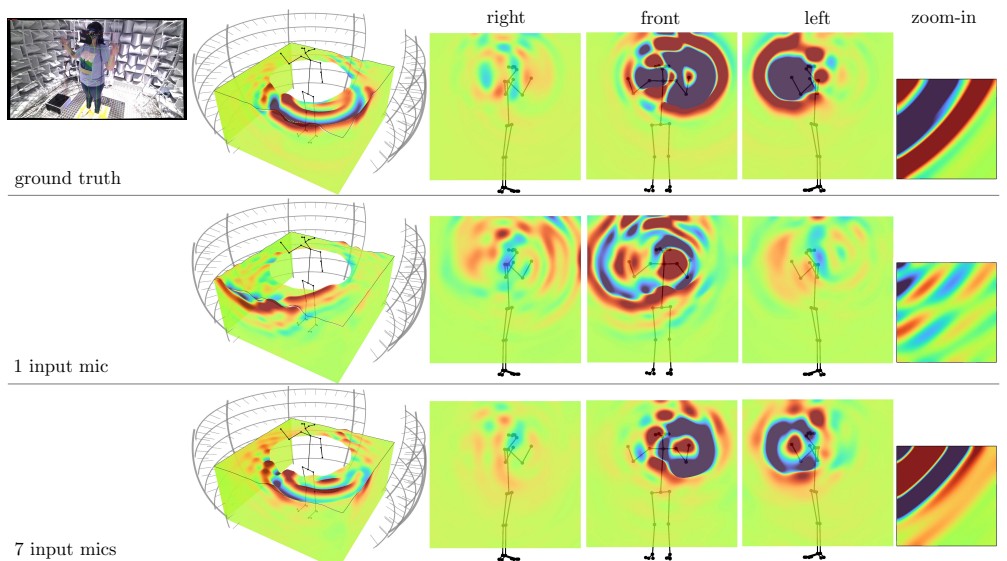

Figure 6: Body pose alone is not always enough to disambiguate the correct origin of sound. Using multiple microphones helps infer the missing spatial information.

dataset of its kind that contains multimodal data of speech and body sounds, which paves the way for a more comprehensive exploration of modeling the 3D spatial sound of humans.

**Implementation Details.** In the encoders, the pose is mapped onto a 32-d embedding $f_{\mathbf{p}}$, while the audio input $\mathbf{a}_{1:T}$ is mapped to 128-d features $f_{\mathbf{a}}$ after time-warping. For the audio decoder, we use $M = 10$ blocks, in which each Conv1D layer has 128 channels and kernel size 3, and the dilated Conv1D layer starts with a dilation size of 1 and doubles the size after each block. To transform the output features from 128 channels to a single channel audio, we employ two sequential Conv1D layers with ReLU activations. During training, we fetch 4 audio clips from the training set as a batch and randomly select 38 target microphones from the 345 available target microphones in the capture stage, *i.e.*, a total of 152 1s audio clips are processed in each forward pass. We use Adam to train our model for 125 epochs with learning rate 0.002. When training with shift-$\ell_2$ and multiscale STFT, we multiply the multiscale STFT loss with weight 100. All experiments are run on 4 NVIDIA Tesla A100 GPUs and each needs about 48 hours to finish.

**Evaluation Metrics.** We report results on three metrics, signal-distortion-ratio (SDR), $\ell_2$ error on the amplitude spectrogram, and angular error of the phase spectrogram. While SDR measures the quality of the signal as a whole, amplitude and phase errors show a decomposition of the signal into two components. High amplitude error indicates a mismatch in energy compared to the ground truth, while high phase errors particularly occur when the delays of the spatialized signal are wrong. Note that the angular phase error has an upper bound of $\pi/2 \approx 1.571$. We report amplitude errors amplified by a factor of 1000 to remove leading zeros.

## 5.1 The Role of Body Pose

The ablation of pose information is studied as shown in Table 1. We use all seven head-mounted microphones as input and vary the pose information, removing all the poses or only using the head pose. As mentioned in Section 3, the non-speech data is significantly different from the speech data,

Table 3: Loss ablation. The proposed shift-$\ell_2$ loss in combination with a multiscale STFT loss outperforms other losses on most metrics.

| | non-speech | | | speech | | |
|---|---|---|---|---|---|---|
| loss | SDR ↑ | amplitude ↓ | phase ↓ | SDR ↑ | amplitude ↓ | phase ↓ |
| $\ell_2$ | 2.557 | 1.064 | 0.449 | 7.610 | 10.594 | **1.093** |
| multiscale STFT | −2.222 | 0.797 | 0.557 | −5.077 | **6.898** | 1.902 |
| multiscale STFT + $\ell_2$ | 2.394 | 0.936 | 0.465 | 8.566 | 8.477 | 1.204 |
| shift-$\ell_2$ | 2.956 | 0.910 | **0.447** | 8.899 | 9.716 | 1.136 |
| shift-$\ell_2$ + mult. STFT | **3.004** | **0.784** | 0.454 | **9.580** | 7.059 | 1.184 |

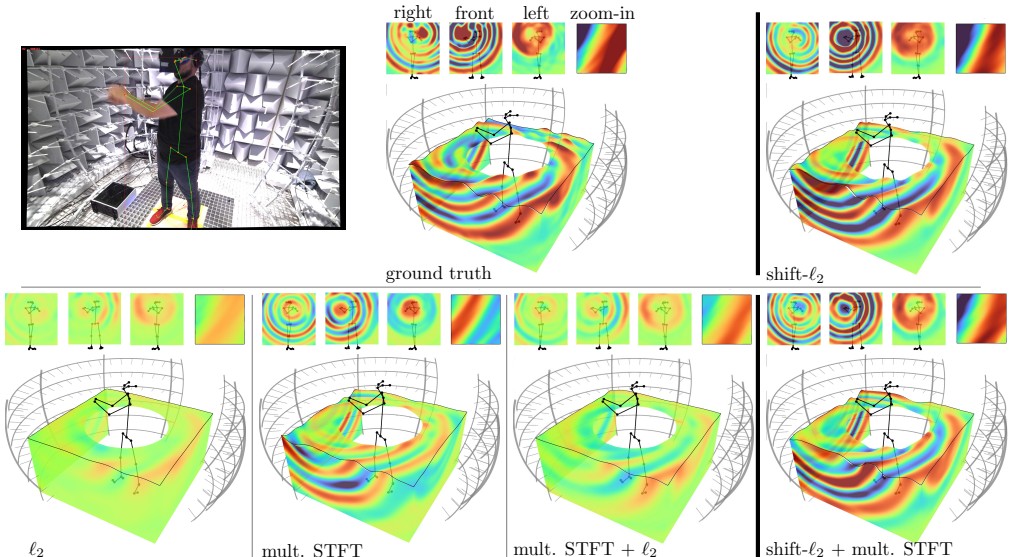

Figure 7: While commonly used losses like $\ell_2$ and multiscale STFT underestimate the sound field energy and have spatial alignment errors, the proposed shift-$\ell_2$ loss enables accurate sound field reconstruction. Fine-grained details further improve when paired with a multiscale STFT loss.

and thus we report quantitative results on speech and non-speech data separately. In Table 1, our model with full body pose outperforms variants with no pose or headpose only on all three metrics because full body pose provides important position information of sound sources. Compared to the no pose version, the headpose leads to a huge improvement in speech data since the headpose itself is capable of providing necessary position information on the speech case.

Figure 5 illustrates how these quantitative differences manifest in the sound field. In the case of a finger snap above the head, as shown in the figure, the sound source is estimated at the center of the 3D space if no pose information is available. With only headpose, the model puts all sound sources in the head region where speech would be produced. Finally, when provided with full body pose, the model is able to correctly locate the snap at the right hand location above the head.

## 5.2 The Role of Head-Mounted Microphones

We also study the impact brought by the number of input microphones. Table 2 lists the results of our model trained with 1, 3, and 7 head-mounted input microphones respectively. As can be observed, our model with 3 microphones shows a clear improvement over that with only one input microphone since, complementary to pose information, some spatial information can be inferred from a 3-channel input audio but monaural audio implies nothing about the space. Compared to 3 input mics, 7 microphones can further boost the overall performance.

Figure 6 illustrates a challenging example where body motion alone does not give a clue whether the finger snap originates from the left or the right hand. With a single microphone as input, the model can only guess, and in the example shown in the figure, it places the origin of sound in the wrong hand. In contrast, using 7 microphones as input, the model is able to estimate the direction of arrival of the sound and, consequently, place its origin in the correct hand.

Table 4: Our method outperforms baselines both quantitatively and qualitatively.

| methods | non-speech | | | speech | | |
|---|---|---|---|---|---|---|
| | SDR $\uparrow$ | amplitude $\downarrow$ | phase $\downarrow$ | SDR $\uparrow$ | amplitude $\downarrow$ | phase $\downarrow$ |
| Only time warping | −0.016 | 1.031 | 0.502 | −4.217 | 20.675 | 1.532 |
| Ours w/o time warping | 2.671 | 0.808 | 0.461 | 8.986 | 7.691 | 1.239 |
| Ours | **3.004** | **0.784** | **0.454** | **9.580** | **7.059** | **1.184** |

## 5.3 Accurate Sound Field Learning with Shift-$\ell_2$

In order to evaluate the effectiveness of our new loss function proposed in Section 4.3, we train our model with different loss functions. As shown in Table 3, we compare the new loss function shift-$\ell_2$ with traditional losses like $\ell_2$ or multiscale STFT Yamamoto et al. [2021]. Although not always the best on all three metrics, this new loss empowers our model to achieve a more balanced performance, unlike traditional loss functions. Moreover, if combined with multiscale STFT, it can further improve the SDR value while lowering the amplitude error, on both speech and non-speech data.

A qualitative evaluation of the different losses in Figure 7 shows the deficiencies of $\ell_2$-loss and multiscale STFT loss. With darker colors (red and blue) representing high energy and green representing low energy, both losses severely underestimate the energy in the predicted signal. The proposed shift-$\ell_2$ loss, on the contrary, matches the ground truth signal well. When combined with a multiscale STFT loss, we observe even more fidelity and better sound field reconstruction in the fine grained details, see the zoomed-in example for shift-$\ell_2$ + multiscale STFT in Figure 7.

## 5.4 Comparison to baselines

Being the first to address the problem, there is no existing work that can serve as a meaningful baseline. Existing approaches to sound spatialization Savioja et al. [1999], Morgado et al. [2018], Richard et al. [2021] rely on knowledge of the sound source location or are limited to first-order ambisonic signals – recall that we model 17-th order signals for high enough spatial resolution. We therefore compare our approach to a naive baseline, where the input signal is naively time-warped from the head position towards the receiver microphone on the sphere, with distance attenuation applied as well. Additionally, we run our system without time-warping to demonstrate the impact of time-warping. As shown in Table 4, on all three metrics, our model demonstrates substantial improvements over the baseline. The results from our model without time-warping are inferior to the model with time-warping.

Additional visualization examples, including failure cases, can be found in the supplemental material.

## 6 Conclusions

Our pose-guided sound rendering system is the first approach of its kind that can successfully render 3D sound fields produced by human bodies, and therefore enriches recent progress in visual body models by its acoustic counterpart. Enabling this multimodal experience is critical to achieving immersion and realism in 3D applications such as social VR.

**Limitations.** While the results of our approach are promising, it still comes with considerable limitations that require deeper investigation in future work. First, the approach is limited to rendering sound fields of human bodies alone. General social interactions have frequent interactions of humans and objects, which affect sound propagation and the characteristics of the sound itself. Moreover, as the system is built upon an ambisonic sound field representation, it produces accurate results in the far field, yet fails in the near field when entering a trust region too close to the body that is modeled. Generally, the model assumes free-field propagation around the body, which is violated if two bodies are close to each other and cause interactions and diffractions in the sound field. Lastly, the system relies on powerful GPUs to render the sound field efficiently, and is not yet efficient enough to run on commodity hardware with low computing resources such as a consumer VR device.

Nevertheless, correctly modeling 3D spatial audio is a necessary step for building truly immersive virtual humans and our approach is a critical step towards this goal.

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
