# OpenReview forum: "Sounding Bodies: Modeling 3D Spatial Sound of Humans Using Body Pose and Audio"
_NeurIPS.cc/2023/Conference — NeurIPS 2023 spotlight_

### Official Review · Reviewer_R8ko · 2023-06-17

**Soundness:** 3 good
**Presentation:** 4 excellent
**Contribution:** 3 good
**Rating:** 7
**Confidence:** 4

**Summary:**

This paper deals with the problem of predicting sound fields around human bodies and proposes a method that exploits binaural audio signals and human body motions. The proposed method first encodes binaural audio signals and human poses and then decodes them into audio signals that are supposed to be captured by surrounding microphones. Finally, the proposed method renders the predicted signals into sound fields. The proposed method also introduces a loss function that is robust against small time shifts of time-domain audio signals. Experimental evaluations with originally collected datasets demonstrate that the proposed method captured the ground-truth sound fields reasonably well and it requires full body poses instead of only head poses for obtaining good predictions.

**Strengths:**

1. The problem dealt with in this paper is novel as far as I know. Although sound field prediction is one of the standard tasks in audio signal processing, sound field prediction with blindly moving microphones would be novel and challenging. This paper solves this problem by (1) assuming that sound is emitted from a human and the human wears microphones and (2) integrating pose estimation of the target human.

2. The proposed method is technically sound. Since the current problem setting is sensitive to time shifts, the proposed method handles audio signals in the time domain, which is not so popular in multi-modal and cross-modal analysis with deep learning. The proposed method achieves time-domain processing by incorporating WaveNet-like decoders.

3. The newly introduced loss function is also technically sound. Time-domain signals are more sensitive to time shifts than frequency-domain ones. The proposed loss function named the shift-\ell_2 loss effectively mitigates this problem while maintaining its discriminability.

4. The experimental evaluations demonstrate that
(1) the proposed method reasonably worked well,
(2) full-body poses contributed to the performance improvement of sound source localization compared with head poses, and
(3) the proposed loss function was effective for non-speech sounds.

**Weaknesses:**

1. The current experimental evaluations do not contain comparisons with other existing methods. I understand that the current problem setting is novel and thus there are no existing methods that can be directly applied to this problem. However, several previous methods for sound field prediction have already been known, and experimental comparisons with one of those methods will be informative for readers.

2. The problem setting seems to be too extreme. Sound field prediction and reconstruction are one of significant problems in audio signal processing. However, the current problem setting requires (1) humans as sound sources, (2) microphones worn by humans, and (3) external video cameras. In particular, the first constraint seems to be too strict. Some justifications and potential applications will be required.

**Questions:**

None.

**Limitations:**

I could not find any negative societal impacts of this work.

---

> ### Author Rebuttal · Authors · 2023-08-09
>
> 1. Comparison with previous methods for sound field prediction
>
> We recognize that a comparison with other methods would be beneficial. However, as pointed out by the reviewer as well, the sound field reconstruction from egocentric data is a novel problem and no existing methods can be directly applied to this setting. Moreover, we are not aware of any sound field prediction methods that can be adapted in a straightforward fashion without involving substantial modifications. If the reviewer has any specific papers in mind, we are happy to consider them for the final manuscript.
>
> 2. Problem setting – justifications and potential applications
>
> We are considering VR telepresence application where people interact in VR space as full-body avatars. In this application the constraint 2 (microphones worn by humans) is easily satisfied as it is common for VR headsets to employ microphone arrays. Regarding the constraint 1 (humans as sound sources): since this is the first work to address the problem of sound field prediction from egocentric data and considering that humans are main sources of sound in a telepresence system, we decided to focus on speech and body sounds in the current work. We plan to extend the work to human-object interaction in future. Finally, regarding the constraint 3 (external video cameras), the proposed method requires pose as input, which can be obtained from external cameras, or from the headset itself (note: the current generation of headsets such as Meta Quest 2 are able to track the upper body / hands while the lower body remains a challenge). We will more clearly highlight this application scenario in the final manuscript.

---

> ### Comment · Reviewer_R8ko · 2023-08-14
> **I have read the auhor response.**
>
> Just an acknowledgment that I have read the author response. Although the lack of reasonable baselines is still a weakness of this paper, I think that it is not a serious issue for considering the acceptance/rejection of this paper.

---

### Official Review · Reviewer_k1Fj · 2023-07-05

**Soundness:** 4 excellent
**Presentation:** 3 good
**Contribution:** 3 good
**Rating:** 7
**Confidence:** 4

**Summary:**

The paper introduces a novel model for a novel task to render a 3D spatial sound from human body motion and audio collected by headset microphones. The authors also present a novel dataset containing human body motion and audio for this task. The model takes encoded audio, pose features, and target microphone position to render the corresponding spatial sound. A novel shifted-l2 loss is also proposed to address the issue of big spikes in l2 error due to the small shifts of impulsive signals. Experiments and results demonstrate the effectiveness of the proposed method.

**Strengths:**

1. The motivation and objectives are clear. To my best knowledge, it is indeed the first paper that aims to render 3D spatial sound from a human body pose.

2. The dataset itself is valuable. It could unlock various interesting research projects in the future.

3. The system is simple but effective. The proposed shifted-l2 loss is interesting and could be useful in other audio applications as well.

4. Analyses of the roles of body pose and head-mounted microphones are well-done in the ablation studies.

5. The phase is important and I am glad that the authors consider it in the optimization.

**Weaknesses:**

1. While extensive quantitative evaluations and analyses are done, it is often challenging to understand the gap in the perceptual level. It would be great if there is a subjective evaluation to demonstrate both the importance of the proposed task/approach and the contribution of the technical components.

2. It would also be nice to have more samples in the supplementary material for a subjective evaluation of the proposed approach.

3. I wonder whether any failure cases are appearing in the current exploration stage.

4. audio and visual data time synchronization is often challenging but the details to achieve that are missing.

5. As the authors mentioned in the limitation, the current approach could not handle spatial sound for human-object interaction, mainly for far-field modeling, and require intensive computation resources.

**Questions:**

1. Why do you use only 1s audio clips during training? Is there a bottleneck that avoids having a longer audio sequence during training? Or is it simply because of the high sampling rate (48khz)?

2. Would it be possible to have data with multi-person and human-object interaction in the future? What might be the challenges to record these data?

3. I am interested in the hyper-parameter choices for the shifted-l2 loss. How is the L=128 varied if the sampling rate differs?

---

> ### Author Rebuttal · Authors · 2023-08-09
>
> Weaknesses:
>
> 1. Subjective evaluation
>
> We agree that subjective evaluations are important. At the same time, the proposed work deals with sound spatialization, and it is not straightforward to perceptually evaluate spatialization quality. Study participants should be able to see the scene in 3D and an effective way to evaluate the correctness of the sound spatialization should be devised. To our knowledge there is no standard procedure for this kind of evaluations. Additionally, complex user studies in VR are not repeatable by the research community and therefore add limited value for other researchers.
>
> 2. & 3. Additional samples + failure cases
>
> We agree that additional samples could be useful for the readers. We will include more examples in the supplementary material of the final manuscript, also showcasing the cases where the model does not perform well.
>
> 4. Audio-visual time synchronization
>
> The audio system is a distributed array of MADI-based analog-to-digital converters. Each analog-to-digital converter is phase-locked to each other via the same WordClock signal. The video system is a distributed array of Kinect Azure cameras. Each Kinect is set to “subordinate mode” where the unit will only take an image when they get a digital high signal from the sync-in port. The Kinect array is in a “star” pattern where one master unit sends a digital high signal out to a distribution amplifier. The master Kinect digital high signal is then propagated to all of the subordinate Kinects. We also record the digital high signal into our analog-to-digital audio converters so we have correspondence between a visual frame and an audio sample. We will include these technical details in the supplementary material of the final manuscript.
>
> Questions:
>
> 1. Why 1 second duration?
>
> Because of the high sampling rate (48 kHz) and the fact that for each input sequence (1-7 headset mics) we predict multiple output signals (38 randomly selected mics from the capture stage), there is a memory bottleneck that prohibits using longer sequences. One-second-long sequences worked fine in our case. We would also like to point out that since the receptive field of the network is 2047 samples, there are no benefits of using a longer sequence.
>
> 2. multi-person and human-object interaction data?
>
> We plan to extend the proposed framework to human-object interaction by collecting a new dataset consisting of people interacting with different kinds of objects. At least in theory, no network architecture or capture system changes are needed for this extension. On the other hand, multi-person interaction is more challenging: no information about the sound field in-between the people would be available since the microphones surround the scene. Adding additional microphones in-between the participants would be challenging and would significantly restrict the body movement.
>
> 3. Hyper-parameter choice: why L = 128 and its dependence on sampling frequency?
>
> Given the size of the capture dome (diameter = ~2.44m, microphone tip-to-tip) there exist an upper limit of the sound propagation delay that can occur in this setting between two extreme positions in the dome, which is around 340 samples (48000 * 2.44 / 343). Since the network deals with the time-differences-of-arrival between the input headset microphones and output capture dome microphones, the actual upper limit that the network needs to account for is roughly the half, i.e., ~170 samples. We choose 128 as the closest power of two.

---

> > ### Comment · Reviewer_k1Fj · 2023-08-17
> > **Post rebuttal**
> >
> > Thank the authors for the responses. Most of my questions are well addressed. Since the task is novel, I believe it makes sense that the initial settings are relatively simple to better understand the problem itself. I also appreciate that the proposed method is not overcomplicated. Thereby, I will keep my original rating and prefer to accept the paper.

---

### Official Review · Reviewer_of5R · 2023-07-06

**Soundness:** 4 excellent
**Presentation:** 4 excellent
**Contribution:** 4 excellent
**Rating:** 8
**Confidence:** 4

**Summary:**

This work introduces an approach for generating spatial audio from a 3D human pose and microphones placed close to the subject’s head, e.g. on a VR/AR headset, as is the case here. The authors first collect a multimodal dataset that contains 3D bodies and audio, recorded using multiple Kinects for body tracking and 345 microphones for audio acquisition. The dataset contains multiple participants in different outfits and poses, generating different sounds following a pre-defined script. To generate spatial audio, the authors propose a model that receives as input the audio recorded from the headset microphones and the 3D body pose and predicts the audio signal recorded at a microphone of the spherical capture array. Using the audio at each microphone and the locations of each microphone, the authors compute harmonic sound field coefficients that best represent the sound field.

**Strengths:**

- Important problem with large impact: The authors tackle an important problem for AR/VR applications that has not yet received a lot of attention. Generating accurate sound fields promises to increase immersion and the realism of virtual experiences. The proposed dataset will be an important benchmark for the community and open up new and exciting avenues of research.
- Novelty: The proposed dataset is an important contribution with unique data that are not publicly available.
- A loss that is well-motivated by prior-work and adapted to the problem structure that leads to clear benefits.
- A simple model that merges different types of information, using well-proven components.
- Writing: The paper is well-written and easy to follow.
- Experiments: The experimental analysis is clear, highlights the effects of each design choice and provides insights into what makes the method work.


**Weaknesses:**

- The related work section lacks a discussion of recent methods that model sound fields using neural network fields, such as:
  - [Learning Neural Acoustic Fields](https://www.andrew.cmu.edu/user/afluo/Neural_Acoustic_Fields/).
  - [INRAS: Implicit Neural Representation for Audio Scenes](https://openreview.net/forum?id=7KBzV5IL7W)

These could also be useful baselines for the model, both in terms of accuracy and in terms of computation compared to the proposed method. Furthermore, the following datasets could be a useful discussion point for the related work section:
  - SoundSpaces: Audio-Visual Navigation in 3D Environments: A discussion of whether a similar dataset as the one proposed in this work could be constructed by simulating audio propagation from artificial bodies.
  - Talking With Hands 16.2M: A Large-Scale Dataset of Synchronized Body-Finger Motion and Audio for Conversational Motion Analysis and Synthesis: I believe the related work section could benefit from a comparison with this paper in terms of granularity of pose representation and sound field, number of persons and type of conversations.


**Questions:**

- Do you use only body keypoints? Or do you include hand keypoints from OpenPose?
- I believe that a comment or discussion on the granularity of the representation of the human body would be useful. The ablation clearly shows that a more detailed representation of the human body helps improve the quality of the predicted sound field. I would like to hear the authors' answer on the following questions:
  - Is the robustness of the estimated keypoints important? Do noisy keypoints significantly affect the quality of the sound field?
  - Is the lower body important for sound quality?
  - Would a finer representation of the subject help? For example, would including hand or face keypoints from OpenPose help?
Some implementation details that should be clarified:
- How was the size of the window in loss function chosen?
- How was the training sampling strategy selected? Maximization of GPU memory usage?

- Finally, regarding the performance of the model, it would be useful to have a visualization of the distribution of errors around the sphere of microphones. Is there a subset of microphones where the audio signal is better reconstructed?


**Limitations:**

The authors present the limitations of the proposed dataset and method. They acknowledge that the current work is limited to modeling sound effects for a single person in an artificial scene and that it cannot model sound close to the body.

---

> ### Author Rebuttal · Authors · 2023-08-09
>
> Weaknesses:
>
> We thank the reviewer for suggestions that would help make the related works section more comprehensive. We will add a paragraph that discusses mentioned works. NAF and INRAS both leverage implicit representations for sound field modeling, but they focus on room impulse responses and require precise sound source locations. SoundSpaces also focuses on room acoustics while we’re interested in audio propagation around the human body insulated from room reverberation. Besides, SoundSpaces is a synthetic dataset and the fidelity of the sound field may be bounded by the adopted room acoustics modeling algorithm. Talking With Hands 16.2M is a large-scale dataset of body-finger motion and conversational speech. Apart from body poses, the dataset captures finger motion as well. Talking With Hands 16.2M uses only  two directional microphones, which are not enough for sound field modeling.
>
> Questions:
>
> 1. Are only body keypoints used? Would including hand or face keypoints help?
>
> We are using body keypoints only in this work. While we plan to include detailed finger pose as well, this will require improvements to the capture system. We found the quality of finger tracking from the five Kinects used in our setup insufficient to be helpful.
>
> 2. Impact of noisy keypoints & importance of lower body?
>
> In the attached PDF we added an ablation study with noisy keypoints (Gaussian noise with 10cm standard deviation) and without the lower body. As expected, performance degrades in both cases, although the noise has a bigger impact overall, while lower body influence is more marginal.
>
> 3. Size of the window in loss function?
>
> Given the size of the capture dome (diameter = ~2.44m, microphone tip-to-tip) there exist an upper limit of the sound propagation delay that can occur in this setting between two extreme positions in the dome, which is around 340 samples (48000 * 2.44 / 343). Since the network deals with the time-differences-of-arrival between the input headset microphones and output capture dome microphones, the actual upper limit that the network needs to account for is roughly the half, i.e., ~170 samples. We choose 128 as the closest power of two.
>
> 4. Sampling strategy during training – maximization of GPU memory usage?
>
> Yes, the sampling strategy maximizes the GPU memory usage: for each input sequence we randomly select 38 out of 345 microphones on the capture dome; selecting a higher number leads to out-of-memory failures during training.
>
> 5. Spatial distribution of errors?
>
> We added a visualization of the distribution of errors around the sphere of microphones to the attached PDF. All participants were facing mostly towards -25 deg azimuth (opposite to the entrance of the dome) and we notice the best performance is achieved in that region in terms of SDR and phase loss. The amplitude loss shows the opposite pattern, but we note that, unlike phase that does not depend on signal energy and SDR that is a normalized value, amplitude loss may be higher in the frontal region simply because the signal energy for speech is generally higher in the frontal direction.

---

> > ### Comment · Reviewer_of5R · 2023-08-17
> > **Reply to rebuttal**
> >
> > I would like to thank the authors for their reply and for including the requested figures. My questions have all been answered by the reply and the provided rebuttal PDF. I will keep my original rating and recommend accepting the paper.

---

### Official Review · Reviewer_1NH5 · 2023-07-07

**Soundness:** 2 fair
**Presentation:** 3 good
**Contribution:** 2 fair
**Rating:** 5
**Confidence:** 3

**Summary:**

The paper tackles the task spatializing a mixture of speech and body sounds at different points in a sphere around a human body without explicitly recording or knowing the sound source location. Towards that goal, the paper captures a new dataset of humans speaking and making different body sounds in a very controlled and extensive setup. The paper proposes a model that takes as input the time-warped audio recorded by a VR headset and the body pose sequence, and renders the audio at different points on the sphere conditioned on the azimuth and elevation of the points. Furthermore, the paper proposes a novel and intuitloss function to accurately model both speech and short-lived body sounds. The paper compares its model with a heurisitcal baseline, evaluates different versions of the loss function, and also reports results for different microphone channel count and an ablation of the model.


Post author-reviewer discussion: I have read the rebuttal. The responses are detailed and answer my questions. I have increased my score.

**Strengths:**

1. The proposed task is interesting.

2. The proposed loss is intuitive and seems to be well-designed.

3. The dataset could also be a valuable contribution for the community.

**Weaknesses:**

1. a) The dataset capture seems to be done in just 1 environment. There is no discussion or experiments on the model's generalization to other environments.
   b) The capture setup seems to be pretty complicated and could limit the applicability of the model to novel environments if training needs to be redone for new environments.

2. a) The capture scene seem to be simple in the sense that it doesn't contain other objects, is single-room in nature, etc. How would the model fare for more complex realworld scenes?

    b) Would the model be able to accurately render sounds at locations that don't have a path of direct sounds from the source?

3. How is the source location in L150-1 determined?

Minor:
1. The shift-l2 loss doesn't seem to perform by itself in most cases (as also pointed out by the authors) even though the design is pretty intuitive.

**Questions:**

I would like to request the authors to address the questions/concerns I raised in 'Weaknesses', in the rebuttal.

**Limitations:**

The paper discusses its limitations, and I can't think of any serious negative societal implications of the work that need to be discussed in the paper.

---

> ### Author Rebuttal · Authors · 2023-08-09
>
> Questions/concerns:
>
> 1. a) & b) Generalization to other environments
>
> We would like to clarify that we are considering a VR telepresence application where people interact in a VR space as full-body avatars. In this scenario we can distinguish between two environments: (1) the real physical environment the user is in while driving their avatar, and (2) the virtual space where the interaction is taking place.
> For (2), it is desirable to model the spatial body sounds in an anechoic environment, which allows to add arbitrary room reverberations using traditional signal processing methods (e.g., convolution with room impulse responses).
> For (1), the situation is more challenging: a user’s real environment (the “input environment”) can be noisy and reverberant. There is a huge corpus of research around denoising and we are not addressing the problem in this paper. Yet, we did a test to see how our model behaves with inputs from a highly reverberant room. An example is shown in the rebuttal PDF. The example shows that, although the model is trained on anechoic data, the origin of sound is correctly spatialized in the reverberant case as well. However, we note that high levels of undesired reverberation are present, and in general a dereverberation step would be required in this application scenario.
>
> 2. a) Simple scene with no objects
>
> Since this is the first work to address the problem of sound field prediction from egocentric data and considering that humans are main sources of sound in a telepresence system, we decided to focus on speech and body sounds in the current work. We plan to extend the proposed framework to human-object interaction by collecting a new dataset consisting of people interacting with different kinds of objects in future work.
>
> 2. b) Sound field reconstruction inside regions without free-field propagation?
>
> Since the network is trained only on observations that surround the acoustic scene (the microphone array encompasses the capture space), reconstructions inside any region that violates the free-field assumption are not going to be accurate. We mention this as a limitation of the proposed method in the Conclusions section.
>
> 3. How is the source location in L150-1 determined?
>
> Source localization is performed implicitly by the network. We do perform the geometric warping to help the network with the sound propagation delay, but since the origin of sound is unknown, we perform the warping from six body joints that are most frequent sound emitters and feed all warped signals as inputs to the network. More details can be found in the section A Time Warping of supplementary material.

---

> > ### Comment · Reviewer_1NH5 · 2023-08-17
> >
> > The rebuttal responses are detailed and answer my questions. I am increasing my score.

---

### Author Rebuttal · Authors · 2023-08-09

We would like to thank all the reviewers for taking the time to review the paper and their help in improving the quality of the manuscript with their constructive feedback. We replied to each reviewer’s comments and suggestions individually below. Attached to this rebuttal is also a PDF that includes: (1) a figure with examples of model outputs using data from anechoic chamber and a reverberant room; (2) a table with ablation study that shows impact of keypoint robustness and granularity of the representation of the human body; and (3) a figure showing the distribution of errors around the sphere of microphones.

---

### Decision · Program_Chairs · 2023-09-21

**Decision:**

Accept (spotlight)

**Comment:**

This paper received all accept ratings and the provide rebuttal resolved the concerns raised by the reviewers. In particular, the pros include: (1) solving a novel and interesting problem, (2) contributing a new dataset, (3) a new and effective loss, (4) good writing. Given that, I recommend accept for spotlight.